# Impact of 3D radiative transfer on airborne $NO_2$ imaging remote sensing over cities with buildings

Marc Schwaerzel[1,2], Dominik Brunner[1], Fabian Jakub[3], Claudia Emde[3,4], Brigitte Buchmann[1], Alexis Berne[2], and Gerrit Kuhlmann[1]

[1]Empa, Swiss Federal Laboratories for Materials Science and Technology, Dübendorf, Switzerland
[2]Environmental Remote Sensing Laboratory, École Polytechnique Fédérale de Lausanne, Lausanne, Switzerland
[3]Meteorological Institute, Ludwig Maximilian University of Munich, Munich, Germany
[4]German Aerospace Center (DLR), Oberpfaffenhofen, Germany

**Correspondence:** Gerrit Kuhlmann (gerrit.kuhlmann@empa.ch)

**Abstract.** Airborne imaging remote sensing is increasingly used to map the spatial distribution of nitrogen dioxide ($NO_2$) in cities. Despite the small ground-pixel size of the sensors, the measured $NO_2$ distributions are much smoother than one would expect from high-resolution model simulations of $NO_2$ over cities. This could partly be caused by 3D radiative transfer effects due to observation geometry, adjacency effects and effects of buildings. Here, we present a case study of imaging a synthetic $NO_2$ distribution for a district of Zurich using the 3D MYSTIC solver of the libRadtran radiative transfer library. We computed $NO_2$ slant column densities (SCD) using the recently implemented 3D-box air mass factors (3D-box AMF) and a new urban canopy module to account for the effects of buildings. We found that for a single ground pixel (50 m x 50 m) more than 50% of the sensitivity is located outside of the pixel, primarily in the direction of the main optical path between sun, ground pixel, and instrument. Consequently, $NO_2$ SCDs are spatially smoothed, which results in an increase over roads when they are parallel to the optical path and a decrease otherwise. When buildings are included, $NO_2$ SCDs are reduced on average by 5% due to the reduced sensitivity to $NO_2$ in the shadows of the buildings. The effects of buildings also introduce a complex pattern of variability in SCDs that would show up in airborne observations as an additional noise component (about $12\,\mu mol\,m^{-2}$) similar to the magnitude of typical measurement uncertainties. The smearing of the SCDs cannot be corrected using 1D-layer AMFs that assume horizontal homogeneity and thus remains in the final $NO_2$ map. 3D radiative transfer effects by including buildings need to be considered to compute more accurate AMFs and to reduce biases in $NO_2$ vertical columns obtained from high-resolution city-scale $NO_2$ remote sensing.

## 1 Introduction

Nitrogen oxides ($NO_x$ = NO + $NO_2$) are key air pollutants mainly emitted by fuel combustion, traffic, heating systems, industrial facilities and power plants. Their short lifetime and localized sources result in a high spatial and temporal variability especially in urban areas (Berchet et al., 2017). An attractive possibility to create high-resolution maps (<100 m) of the $NO_2$ distribution in cities is to use an airborne imaging spectrometer. This was first demonstrated for measurements from the Airborne Prism Experiment (APEX) of a Swiss-Belgium consortium (Popp et al., 2012; Tack et al., 2017) and the Geostationary

Trace gas and Aerosol Sensor Optimization (GeoTASO) spectrometer (Nowlan et al., 2016) developed in the United States. A comprehensive comparison between four airborne $NO_2$ imaging spectrometers flown over the city of Berlin was performed during the AROMAPEX campaign (Tack et al., 2019), which included APEX (Schaepman et al., 2015), AirMAP (Airborne imaging DOAS instrument for Measurements of Atmospheric Pollution) (Schönhardt et al., 2015), SWING (Small Whiskbroom Imager for atmospheric compositioN monitorinG) (Merlaud et al., 2013) and SBI (Spectrolite Breadboard Instrument) (de Goeij et al., 2017; Vlemmix et al., 2017).

Airborne imaging spectrometers are passive remote sensing instruments from which we can retrieve $NO_2$ slant column densities (SCD) from backscattered solar radiance spectra. SCDs are defined as the integrated trace gas concentrations along the optical path from the sun, through the atmosphere and towards the instrument. The optical path and therefore the SCD depends on illumination conditions, viewing geometry, the scattering and absorption by air molecules, aerosols and clouds as well as the reflectance of the surface. To obtain a quantity that is independent of illumination and viewing conditions, SCDs are divided by air mass factors (AMF) to compute vertical column densities (VCD) defined as the vertically integrated concentrations from the ground to the top of the atmosphere.

$NO_2$ retrieval algorithms applied to airborne instruments have been derived from algorithms developed for satellite instruments with much lower spatial resolution (> 5 km). These algorithms rely on 1D radiative transfer simulations, which reach their limits at high resolution and in the presence of spatially variable properties of the atmosphere and the surface, as they assume horizontal homogeneity of all parameters within the optical path. A strong indication for the importance of 3D radiative transfer effects is that $NO_2$ maps obtained from airborne imaging spectrometers over cities are spatially much smoother than one would expect from the instrument resolution and compared to maps obtained from high-resolution city-scale dispersion models, which show, for example, strong gradients in the $NO_2$ field along major roads (Kuhlmann et al., 2017). A likely explanation is that the measurements are not only sensitive to $NO_2$ in the vertical column above the observed ground pixels but also to the atmosphere surrounding the pixels, which is known as the horizontal smoothing error or the adjacency effect (Cracknell and Varotsos, 2012; Lyapustin and Kaufman, 2001; Richter, 1990). The adjacency effect has been described especially in the context of high-resolution land-surface remote sensing, but it has also been discussed in the context of atmospheric measurements (e.g., Richter, 1990; Minomura et al., 2001; de Graaf et al., 2016). Evidence for 3D radiation effects due to the presence of clouds in the vicinity of an observed ground pixel has recently been reported for $CO_2$ observations from the OCO-2 satellite (Massie et al., 2017). Such effects are expected to become increasingly important with the increasing resolution of satellite observations (Schwaerzel et al., 2020). An additional complexity over cities to be accounted for is the effects of buildings on the photon paths due to multiple reflections and shielding of the main optical path.

The aim of this study is to quantify, for the first time, the impact of these 3D radiative transfer (RT) effects in the presence of spatially variable surface properties and $NO_2$ concentrations over cities on $NO_2$ retrievals from high-resolution airborne imaging spectrometers. The study builds on the work by Schwaerzel et al. (2020), where we highlighted the importance of 3D RT effects on trace gas remote sensing for ground-based and airborne instruments. In this study, we also use the Monte carlo code for the phYSically correct Tracing of photons In Cloudy atmospheres (MYSTIC) solver of the library for Radiative transfer (libRadtran), which is a Monte Carlo solver that traces individual photons in a 3D model grid (Mayer, 2009; Emde

and Mayer, 2007). These photon paths are converted to 3D-box AMFs that describe the sensitivity of an instrument to a trace gas (here $NO_2$) in each grid box (Schwaerzel et al., 2020). To account for the effect of buildings, we additionally implemented a new urban canopy feature that represents the full 3D building structure of a city with assigned optical properties of the different surfaces. 3D radiative transfer calculations in the urban canopy are not new, but so far focused only on applications not related to trace gas observations such as the computation of radiation budgets and broadband landscape imaging (Gastellu-Etchegorry et al., 2015). The present study is the first to use such a model to investigate 3D effects on trace gas retrievals over a city. In order to isolate different effects on 3D-box AMFs and on the spatial smoothing of the information retrieved from an airborne instrument, we use a comparatively simple setup with 3D buildings representative of a district in the city of Zurich with uniform optical properties of roofs, walls and streets. The model, however, is able to describe the optical properties of each single surface separately.

## 2    Methods

### 2.1    The MYSTIC radiative transfer solver

MYSTIC is operated as a RT solver of the libRadtran package ((Mayer and Kylling, 2005; Emde et al., 2016). The MYSTIC RT solver is based on the Monte Carlo principle to calculate different radiative quantities such as irradiance, radiance, absorption, emission, actinic flux, photon's path length and air mass factors (Mayer, 2009; Emde and Mayer, 2007; Schwaerzel et al., 2020). A Monte Carlo based RTM traces individual photons from the source (e.g., sun) to the instrument (e.g., airborne spectrometer) accounting for ground and atmospheric interactions. Photon paths are treated as a combination of random decisions with a given probability distribution for each interaction (e.g., probability distribution of the scattering direction). By tracing several thousands of photons, the averaged photon paths reaching the instrument become representative of the actual photon paths in the atmosphere. To simulate RT quantities such as 3D-box AMFs, photons are traced backwards from the instrument to the sun, which is equivalent to the forward mode but greatly enhances computational efficiency (Marchuk et al., 1980; Emde and Mayer, 2007). MYSTIC divides the atmosphere into 1D vertical layers or 3D grid boxes with different properties and saves the mean photon path length within each layer or box. A plane parallel geometry is used for 3D AMF calculations, while spherical geometry is possible for 1D calculations.

#### 2.1.1    Air mass factors

AMFs are obtained by dividing the mean photon path length in each layer or box by its height. For a layer or box $i$ with given concentration $c_i$ and optical properties, the 1D-layer or 3D-box AMF is given by

$$AMF_i = \frac{SCD_i}{VCD_i} = \frac{\int_{path} c_i dl}{\int_{z_i}^{z_{i+1}} c_i dz} = \frac{\int_{path} dl}{h_i} = \frac{L_i}{h_i} \tag{1}$$

where $L_i$ is the mean optical path in the layer/box $i$ of all photons that reach the instrument and $h_i$ is the height of the layer/box $i$.

To calculate a total AMF that can be used to convert measured SCDs to VCDs, an a priori VCD ($VCD_i$) is needed for every layer/box $i$. For the 1D case, the total AMF is computed as

$$AMF = \frac{\sum_{k=1}^{n_z} AMF_k VCD_k}{\sum_{k=1}^{n_z} VCD_k} = \frac{\sum_{k=1}^{n_z} SCD_k}{\sum_{k=1}^{n_z} VCD_k} \tag{2}$$

where $n_z$ is the number of vertical layers. For the 3D case, the total AMF is computed as

$$AMF = \frac{\sum_{i=1}^{n_x} \sum_{j=1}^{n_y} \sum_{k=1}^{n_z} AMF_{i,j,k} VCD_{i,j,k}}{\sum_{k=1}^{n_z} VCD_k}$$
$$= \frac{\sum_{i=1}^{n_x} \sum_{j=1}^{n_y} \sum_{k=1}^{n_z} SCD_{i,j,k}}{\sum_{k=1}^{n_z} VCD_k} \tag{3}$$

where $n_x$ and $n_y$ are the number of grid cells in $x$ and $y$ direction. In the 1D case, the AMF depends on the shape of the vertical profile of the trace gas (but not on its abundance) (Palmer et al., 2001). In the 3D case, it additionally depends on the horizontal distribution of the trace gas.

### 2.1.2 Urban canopy implementation

The urban canopy was implemented in libRadtran as a triangle mesh, where each triangle can be assigned different optical and physical properties. Information on vertex positions and optical properties are read from an input file that has to be generated from a 3D building data set prior to the simulation. Using a ray tracing algorithm newly integrated into the model, MYSTIC detects if and where a photon hits a triangle. The interaction of the photon with the surface (absorption or reflection) is then simulated using preexisting MYSTIC functions.

To create a triangular mesh for each building, a Python script was written that converts buildings stored in an ESRI shapefile into triangles. A building consists of one or several flat roofs and several vertical walls. To create triangles for the roof, we connect the polygon centroid to each polygon corner. Wall triangles are created by splitting each wall diagonally into two triangles. Each triangle surface carries information about its albedo and skin temperature. The skin temperatures are used for thermal simulations, a feature that is not used in this publication. The triangular mesh is stored in a NetCDF file readable by MYSTIC. An example file layout is shown in the supplement. The file contains the variable `vertices` (shape: $N_v \times 3$, type: double), which is a list of $x$, $y$ and $z$ coordinates. A mesh of $N_t$ triangles is built from these vertices using the `triangles` variable (shape: $N_t \times 3$, type: int) by storing the indices of the 3 vertices that create the triangles. The variable `materials_of_triangles` (shape: $N_t$, type: int) is used to assign each triangle the index of a material type. The material types are defined using the variables `material_type` (shape: $N_m$, type: string), `material_albedo` (shape: $N_m$, type: double) and `temperature_of_triangle` (shape: $N_m$, type: double) to assign a name, albedo and temperature to each material.

In MYSTIC, each photon is traced step-wise along its optical path from one interaction to the other. To interact with the urban canopy, a ray tracing code searches for hits with one of the triangles during each step. We use the Star-3D library (https://gitlab.com/meso-star/star-3d) (Villefranque et al., 2019), which is a convenient wrapper for Intel® Embree (www.embree.org) to facilitate efficient ray / triangle intersection tests using a bounding volume hierarchy (BVH).

In the case that a ray hits a surface, a Lambertian reflection happens and a new direction is attributed to the photon, which will continue its path until its next interaction. Absorption on surfaces is accounted for by reducing a photon weight by $1 - albedo$. Currently, the urban canopy module only supports Lambertian reflections but an extension, e.g., to specular reflections would be straightforward. For an even more realistic description of the surface reflectance for real applications, BRDF functions would be a well suited method.

## 2.2 Study case in Zurich

To study radiative transfer effects on airborne measurements for a realistic scene, we selected a 1 km x 1 km region in Zurich, Switzerland (see Fig. 1a). The low-rise buildings (10-15 m) with simple geometries are rather typical for Swiss cities. The scene includes roads of different widths and orientations and two open areas without buildings.

### 2.2.1 Buildings

We used 3D building data from a shape file obtained from the Swiss Federal Office of Topography (swisstopo). Each building is defined as a polygon with $x$ and $y$-coordinate for the edges and the centroid (Swiss LV03 coordinates system) with a single height. As described above, the data was converted to a triangular mesh and saved to a netCDF file. For simplicity, we either applied albedos of 0.1 for walls (identical to the used ground albedo) and 0.2 for roofs or used the same albedo of 0.1 for all surfaces (albedos chosen after Mussetti et al., 2020). Figure 1b shows the buildings colored by their heights in the selected model domain.

### 2.2.2 Synthetic $NO_2$ field

We created a simple but quite realistic 3D $NO_2$ concentration field based on the traffic emission inventory of the city of Zurich (see Table 1 in Berchet et al., 2017). For this purpose, the road emissions available as line sources were rasterized at 5 m x 5 m spatial resolution and normalized by the maximum value in the rasterized field. The normalized field was then multiplied with a $NO_2$ concentration of $110 \, \mu g \, m^{-3}$, which is a typical high concentration measured next to busy streets in Zurich (Bär, 2016). A background $NO_2$ concentration of $15 \, \mu g m^{-3}$ was added using a typical low value observed in Zurich (Bär, 2016) (see supplement for details). The maximum concentration is thus $125 \, \mu g \, m^{-3}$ or $2.7 \, \mu mol \, m^{-3}$. Finally, we smoothed the concentration field with a Gaussian filter with a standard deviation of 5 m to mimic the effect of turbulent dispersion. The synthetic field of the $NO_2$ near surface concentrations is shown in Fig. 2a binned on a 50 m x 50 m grid, which is the typical resolution of trace gas measurements from an airborne instrument like APEX.

The vertical distribution was modelled as a linear decrease from the surface value to the background value of $0.65 \, \mu mol \, m^{-3}$ at 100 m. Above 100 m, all profiles decrease exponentially with altitude with an e-folding vertical length scale of 720 m, which was obtained by fitting a function to a measured vertical profile (see supplement). Figure 2b shows a $NO_2$ background profile (blue) and a $NO_2$ profile over the road (red). Figure 2c shows the total VCDs over the selected region in Zurich calculated from the 3D $NO_2$ concentration field. For more details on the generation of the 3D $NO_2$ field, please refer to the supplement.

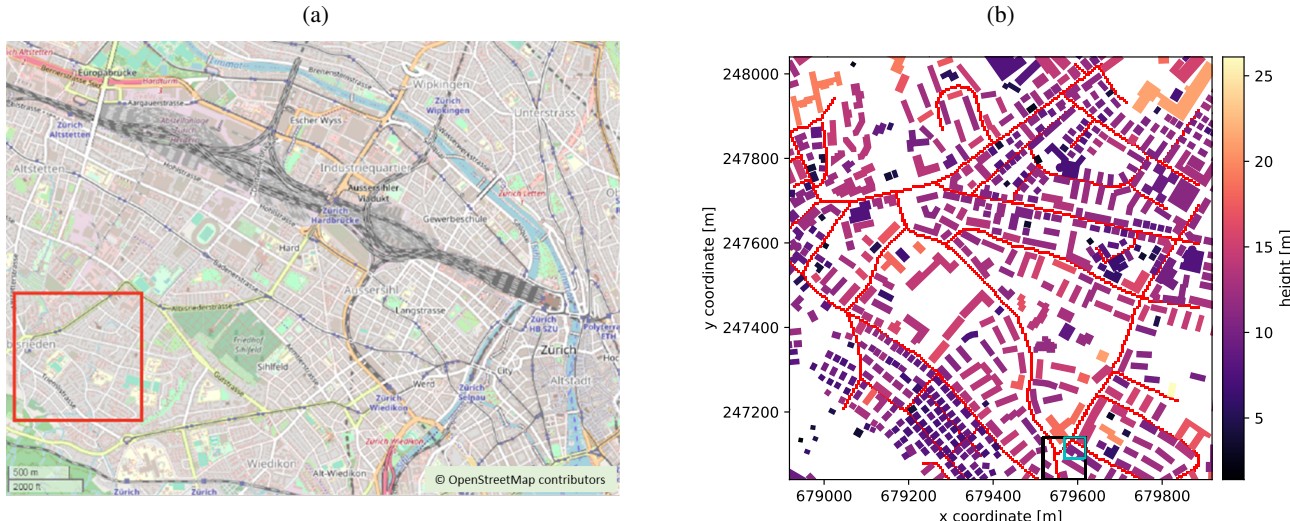

**Figure 1.** (a) Map of Zurich with the location of the study area in red. Map informations were obtained from OpenStreetMap (© OpenStreetMap contributors 2021. Distributed under the Open Data Commons Open Database License (ODbL) v1.0.) (b) Building heights and major roads in the study area with the lower left pixel at x = 678918 m and y = 247040 m in the Swiss LV03 coordinate system. The black square represents the location of the sub-domain and the cyan square the location of a single pixel observation both used in Section 3.3.

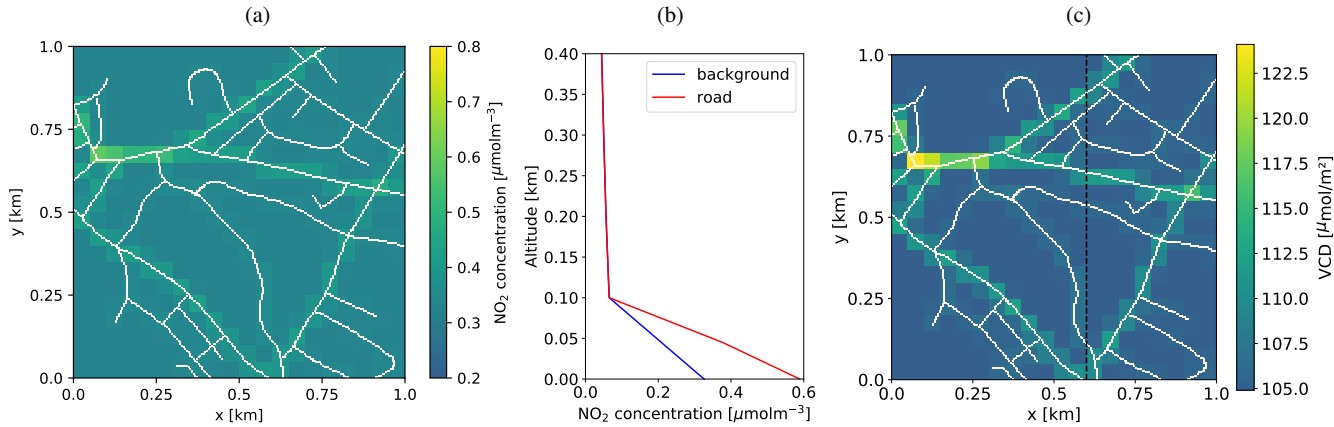

**Figure 2.** (a) $NO_2$ ground concentration map. (b) $NO_2$ vertical profile at a background location (blue line) and over a road at x = 5 m, y = 55 m (red line). (c) Field of vertically integrated $NO_2$ column densities (VCD) with aircraft flight track overlaid as dashed black line.

### 2.2.3 MYSTIC simulations

Our scenario corresponds to an airborne imaging spectrometer flying across the model domain from south to north ($y$-direction) slightly to the east of the center ($x$=600 m) at an altitude of 6 km, which represents the flight altitude of an airborne spectrometer (dashed black line in Fig. 2c). The viewing zenith angle was varied in discrete steps to cover the whole domain in across-flight direction (i.e. in east-west direction). For simplicity, the sun was placed at an azimuth angle (SAA) of either 90° (i.e. west) or 0° (i.e. south). The solar zenith angle (SZA) was set to 60°, which corresponds to values in the morning or afternoon in summer over Zurich. Simulations with a SZA of 30°, which correspond to typical summer noon measurements over Zurich are shown in the supplement.

The simulations were conducted for a standard atmosphere and for a wavelength of 490 nm, which is the center of the $NO_2$ fitting window used for the APEX instrument (Kuhlmann et al., 2016). $NO_2$ absorption features at shorter wavelengths typically used in satellite retrievals are less suitable for APEX due to the high instrument noise at those wavelengths. Note that 3D effects would be stronger at shorter wavelengths due to enhanced Rayleigh scattering. Aerosols were only included in the simulations as a case study to analyse the footprint of the spectrometer (see Sect. 2.3). Table S1 in the supplement provides an overview of the MYSTIC input parameters used in the simulations.

To resolve the spatial variability of surfaces and elevations within each 50 m x 50 m ground pixel, we specified an instrument opening zenith angle corresponding to the 50 m pixel size in $x$-direction and moved the aircraft in 10 discrete steps of 5 m along the $y$-direction. The 10 different 3D-box AMF fields computed in this way were then averaged to a single field per pixel. The 3D-box AMFs were then used together with the synthetic $NO_2$ field to compute total SCDs that would be observed from an airborne instrument (see numerator in Eq. 3). Then, the total AMFs were calculated dividing the total SCDs by the total VCDs. In the same way we also computed SCDs ignoring buildings and SCDs based on 1D-layer AMFs (Eq. 2). Differences between the three types of SCDs can be attributed to 3D and building effects. Since in real applications VCDs would be computed from SCDs, we also use the SCD field calculated with 3D-box AMFs including buildings as the "true" SCD measured by the airborne spectrometer and calculate the VCD field using total AMF calculated from 1D-layer AMFs and 3D-box AMFs without considering buildings to show the errors caused by these simplifications.

### 2.3 Footprint of an airborne spectrometer

To study the sensitivity of a single measurement to the surrounding of the ground pixel, we simulated the horizontal distribution of 3D-box AMFs close to the ground for a single observation scenario. The simulations were conducted without aerosols and for a typical urban aerosol scenario with an aerosol optical depth (AOD) of 0.1. For the scenario, the sensor was placed at $x = 675$ m and $y = 75$ m pointing at a ground pixel with the lower left corner located at $x = 650$ m and $y = 50$ m. The corresponding VZA was 0.72° in the center of the pixel and the VAA was 270° (instrument pointing eastwards) (for details see also table S1 in the supplement).

To obtain a map of the close-to-ground $NO_2$ sensitivity of a single ground pixel measured by an airborne imaging spectrometer during a flight overpass, we simulated AMFs with a 5 m x 5 m horizontal resolution and a 5 m vertical resolution in

the lowest 45 m above ground for 10 equally-spaced instrument opening zenith angles corresponding to the 50 m pixel size in x-direction and moved the aircraft in 10 discrete steps of 5 m along the y-direction. We averaged the obtained 100 AMFs fields, integrated them vertically for the first 45 m and scaled the result by the sum of all pixels to finally obtain a 2D map of the fraction of the sensitivity within one 5 m x 5 m grid cell. In the following, this will be called the $NO_2$ footprint of an airborne spectrometer.

## 3 Results

In this section we first show the instrument incoming radiance calculated with MYSTIC over the selected study area. Second, we analyze the 3D-box AMFs for a single $NO_2$ observation both in the vertical and in the horizontal and compare results with and without buildings. Finally, we compare total AMFs for the complete image obtained from an airborne instrument flying over the study domain as illustrated in Fig. 2c between the solution obtained with the 3D and the 1D RTM to illustrate the importance of 3D radiative transfer effects on the SCD measurements.

### 3.1 Incoming radiance

An airborne imaging spectrometer measures radiance from back-scattered and reflected solar irradiance. Figure 3 shows the instrument incoming radiance at 490 nm for the selected region with the urban canopy. Note that for this example the instrument was placed in the center of the domain at an altitude of 6 km observing the scene with a very wide opening angle of about $10°$ to cover the full 1 km x 1 km domain. The sun is located in the west with a SZA of $60°$. The surface reflectance was set to 0.10, while the reflectance of roofs and walls was set to 0.20 in this example.

Since building surfaces have higher reflectance in this example, the bright building roofs clearly stand out. Some bright walls illuminated by the sun can be seen in the east of the domain. Shadows are also clearly visible in the east of the buildings with taller buildings producing longer shadows. This radiance field is closely related to the close-to-ground $NO_2$ sensitivity. $NO_2$ over a bright surface can more easily be detected than $NO_2$ over a dark surface. In case of a pixel covered by both bright and dark surfaces, the retrieved signal will be more strongly affect by $NO_2$ above the bright parts.

### 3.2 3D-box air mass factors for a single observation

#### 3.2.1 Vertical distribution along the main optical path

3D-box AMFs have a distinct 3D distribution for each ground pixel. Figure 4a shows an example of the 3D-box AMFs projected onto a 2D (x-z) plane by integrating in $y$-direction for an instrument pointing almost in nadir direction at the ground pixel centered at $x$=675 m and $y$=75 m (cyan square in Fig. 1b). Since the pixel is partly covered by buildings, 3D-box AMFs close to the ground (0-10 m) are smaller than those above roof level. Note that in this and all the following simulations all surfaces have a constant albedo of 0.1. Most photons follow the geometric path from the sun to a reflection on the ground or the roof and to the instrument as indicated by the high 3D-box AMFs along this path. Figure 4b shows the 1D-layer AMFs obtained

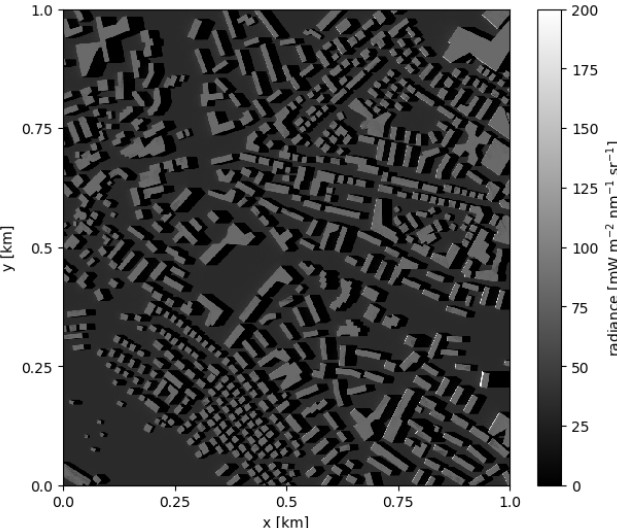

**Figure 3.** Radiance seen by a downward viewing instrument placed in the center of the 1 km x 1 km region at an altitude of 6 km. SZA and SAA are 60° and 90°, respectively.

by integrating the 3D-box AMFs in $x$- and $y$-direction. The decrease close to the ground due to the buildings is clearly visible. Figure 4c shows column AMFs along the x-axis, i.e. 3D-box AMFs integrated in $y$- and $z$-direction. AMFs in the column directly below the instrument are highest, because the collected photons cross at least one of the column boxes when they are scattered into the direction of the instrument. The column AMFs decrease in the $x$-direction with distance to the instrument due to atmospheric absorption and scattering.

Since 3D-box AMFs inform about the sensitivity to $NO_2$, we can conclude that the measurement is mainly sensitive to $NO_2$ along the geometrical optical path. It is not very sensitive to adjacent pixels and also not to $NO_2$ at the surface because of the blocking of the photon path by buildings. The total AMF for the observation presented in this example computed with Eq. 3 is 1.98. The VCD above the ground pixel is 121 $\mu mol\ m^{-2}$ and the SCD computed as VCD times total AMF is 239 $\mu mol\ m^{-2}$.

### 3.2.2 Horizontal footprint

To further illustrate the horizontal sensitivity of an airborne spectrometer to layers close to the ground, Fig. 5 shows the near-surface footprint defined as the vertically integrated 3D-box AMFs from 0-45 m above ground (see Sect. 2.3). The same scene was simulated as in Section 3.2.1 but with a higher horizontal resolution of 5 m x 5 m to illustrate the spatial pattern of the sensitivity in greater detail. The figure is normalized to show the fraction of the sensitivity represented by each 5 m x 5 m pixel.

Figure 5a shows the footprint for a flat surface without buildings without including aerosols. An important part of the sensitivity (51.4%) is located outside the ground pixel. The instrument will thus not only 'see' near-surface $NO_2$ above the ground pixel but also $NO_2$ outside. A major part of the sensitivity is located in the direction of the sun along the main optical

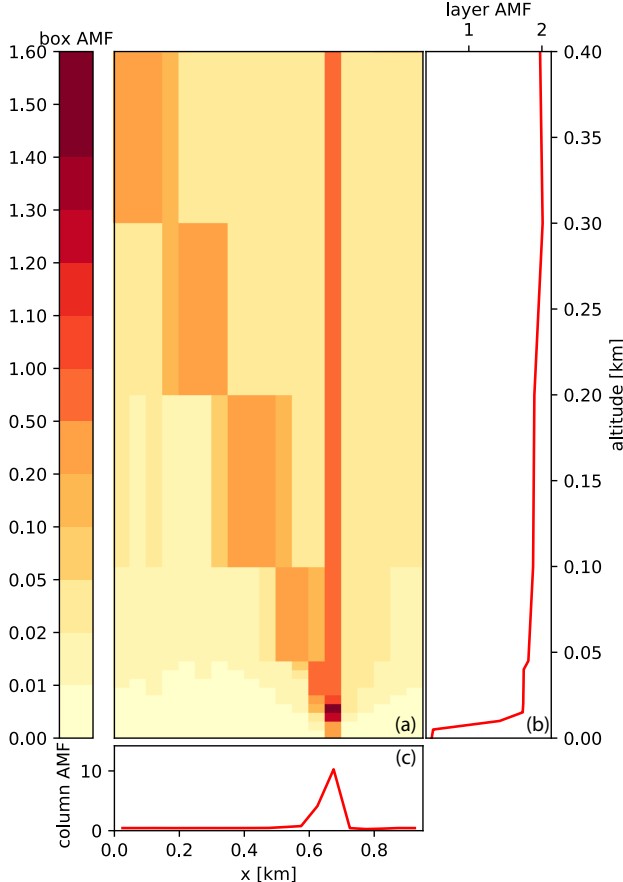

**Figure 4.** (a) Integrated 3D-box AMFs in $y$-direction for a ground pixel at $x$=650 m and $y$=50 m (lower left corner) for an instrument placed at $x$=600 m, $y$=75 m and $z$=6000 m. The sun is at SAA = 90° (west) with a SZA of 60°. (b) Vertical profile of horizontally integrated AMFs (1D-layer AMFs) and (c) Horizontal profile of vertically integrated AMFs (column AMFs).

path. The main optical path is dominated by photons that are either reflected from the ground pixel surface or scattered in the
230 atmosphere above the ground pixel one single time upward into the direction of the instrument. A much smaller fraction is located outside this main path and is caused by photons experiencing at least one more scattering (or reflection) event before being scattered (or reflected) into the main optical path.

Figure 5b presents the same situation but with buildings added to the simulation. Buildings affect the sensitivity both within and outside the ground pixel. The footprint is reduced in the shadows of buildings but may be enhanced over the sunlit sections
of streets due to multiple reflections, as seen near the upper right and lower left corners of the ground pixel. In this example, the part of the sensitivity located outside the ground pixel is 52.6%, comparable to the simulation without buildings.

Figure S7 in the supplement shows the footprint for the aerosol scenario. The aerosol scattering increases the sensitivity contribution from outside the ground pixel to 55% for both the scenario without and with buildings. The contribution from

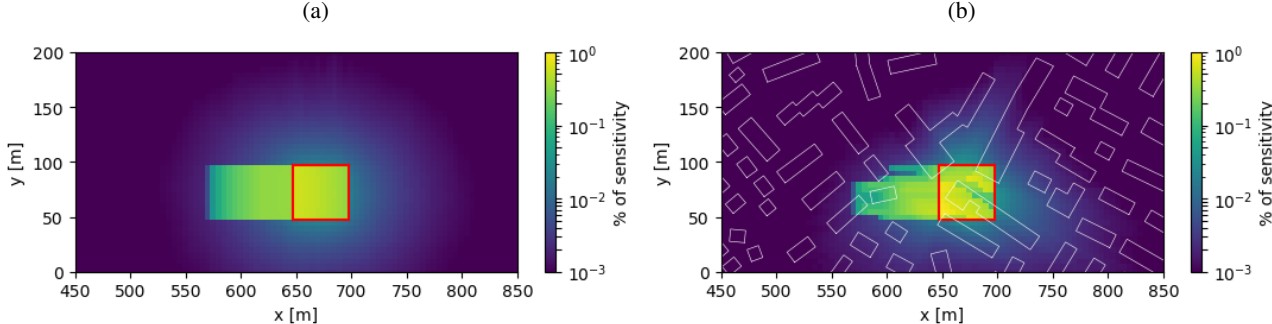

**Figure 5.** (a) Footprint without buildings. 51.4% of the signal is located outside the ground pixel (i.e. outside the red frame). (b) Footprint with buildings. 52.6% of the instrument sensitivity is located outside the ground pixel. Building contours in white.

outside the main optical path is increased to 25% and 32% without and with buildings, respectively, compared to 18% and 24%, when not including aerosols.

The main effect of the 3D optical path of the photons is thus to smear out the sensitivity of a measurement into the direction of the main optical path, which is determined by the viewing and illumination geometry. The presence of buildings further modifies the sensitivity by adding a complex pattern of enhancements and reductions due to the shielding effects of buildings and multiple reflections in street canyons.

### 3.3 Impact of 3D radiative transfer on $NO_2$ imaging over a city

In the previous section we have shown how 3D radiative transfer effects and buildings smear out and modify the sensitivity of a single observation. Here we demonstrate how a complete image of $NO_2$ slant columns is affected by these effects. Note that for simplicity in the following simulations, the ground and building reflectance was set to 0.1.

#### 3.3.1 Effects of 3D radiative transfer and buildings on slant column densities

Figure 6 compares total AMFs computed from 1D-layer AMFs (Eq. 2) with total AMFs computed from 3D-box AMFs (Eq. 3) without and with the urban canopy. The corresponding SCDs are presented in the lower row of the figure, which are related to the total AMFs by a division with VCDs shown in Fig. 2c (see Eq. 1 and Eq. 3). In the following we refer to the SCDs calculated from 1D-layer AMFs as $SCD_{1D}$ and to the SCDs calculated from 3D-box AMFs without and with the urban canopy as $SCD_{3D}$ and $SCD_{3D-UC}$, respectively.

AMFs derived from 1D-layer AMFs (Fig. 6a) are almost horizontally homogeneous. They vary only slightly with the instrument viewing zenith angle and the $NO_2$ distribution. The contribution of the a priori $NO_2$ profile is only significant, when high values collocate with high 1D-layer AMFs values, but as our a priori $NO_2$ profiles only differ in the lower 100 m, the effect is smaller than the noise of the Monte Carlo simulations ($\sigma_{AMF} = 0.014$ and $\sigma_{SCD} = 1.46\,\mu\mathrm{mol\,m^{-2}}$). The corresponding SCDs

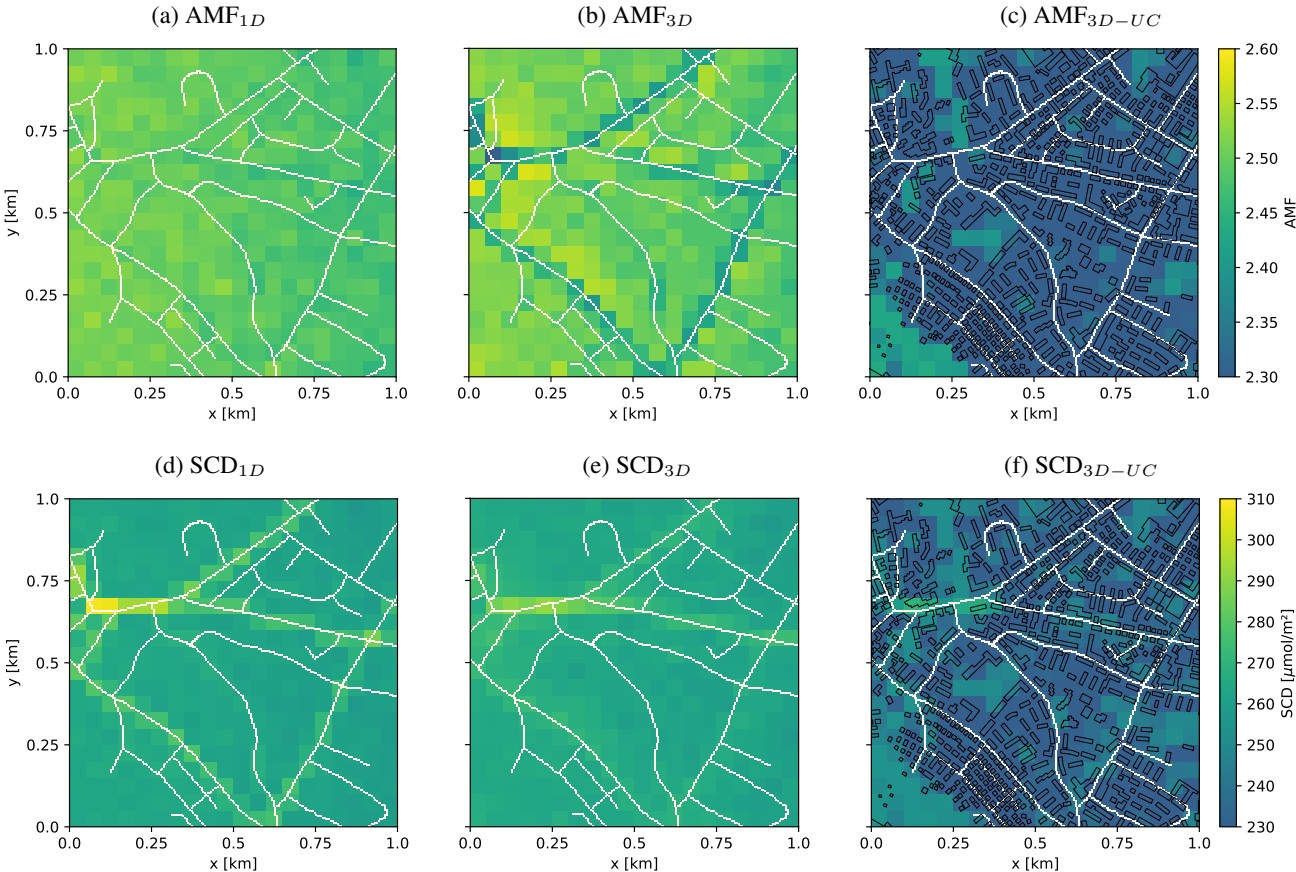

**Figure 6.** AMFs for a simulation with SAA of 90° with 1D-layer AMFs simulation (a), 3D-box AMFs without (b) and with (c) buildings. The respective SCDs are shown in the lower row (d,e,f). Roads are drawn in white and building contours in black for the simulation with buildings.

(Fig. 6d), computed as the sum of the product of the 1D-layer AMFs with the 1D $NO_2$ profile (see Equ. 2) in Fig. 2d show the same pattern as the VCDs with sharply elevated values above roads and a homogeneous background aside. The SCDs are higher than the VCDs because the AMFs are larger than 1.

AMFs calculated with 3D-box AMFs but without buildings (Fig. 6b) are lower over the roads and slightly larger just aside the roads, because the 3D-optical path crosses neighbouring columns with decreased or increased concentrations, respectively. This results in a spatial smearing of SCDs (Fig. 6e) mainly in the direction of the main optical path from the sun in the west, to the ground pixel and to the instrument in the east of the center. Far from the roads, the SCDs are homogeneous and more or less identical to the 1D solution. The difference $SCD_{3D}$ minus $SCD_{1D}$ is presented in Fig. 7. The smearing effect in the 3D solution is visible as negative differences over the roads and positive differences aside especially on the eastern side of

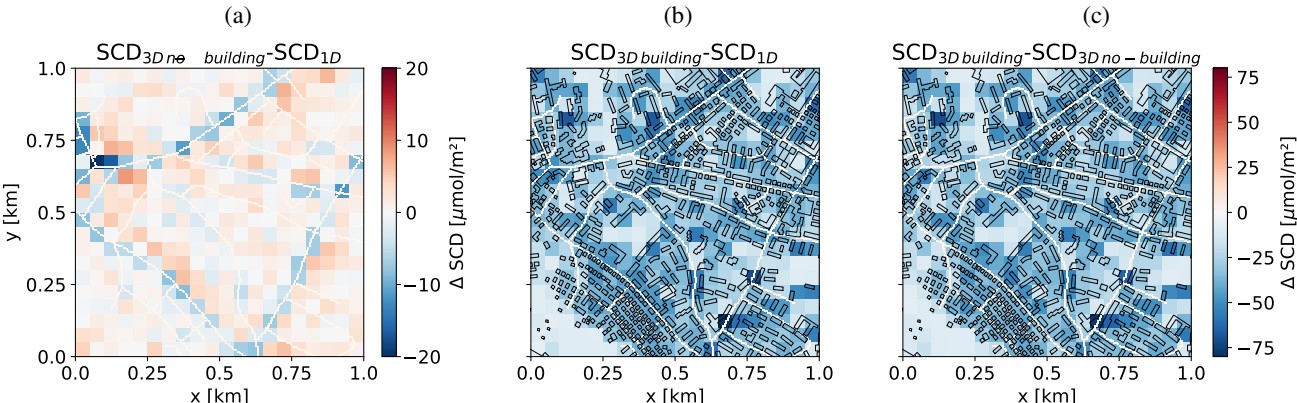

**Figure 7.** (a) Difference plot between SCDs calculates with 3D-box AMFs and the SCDs calculated with 1D-layer AMFs. (b) Difference plot between SCDs calculated with 3D-box AMFs including the urban canopy and SCDs calculated with 1D-layer AMFs. (c) Difference plot between SCDs calculated with 3D-box AMFs with and without including the urban canopy.

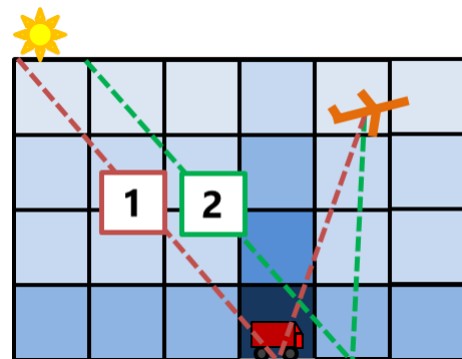

**Figure 8.** Sketch of two main optical paths for an airborne spectrometer pointing at (1) a road in grey and (2) at a pixel aside the road.

the roads opposite to the sun. As a result, the individual roads show up much less prominently in the $SCD_{3D}$ except for roads parallel to the optical path. The reason for this pattern is illustrated in Fig. 8, where the main 3D optical path of the photons
collected when viewing directly at the road (red path 1 in Fig. 8) misses some of the enhanced $NO_2$ concentrations in the elevated levels above the road. In contrast, these enhanced $NO_2$ values are 'seen' by photons collected when viewing to the east of the road (green path 2 in Fig. 8). In the 1D solution, the SCDs are only affected by $NO_2$ in the vertical column directly above the observed ground pixel.

The mean difference between $SCD_{1D}$ and $SCD_{3D}$ is around $1\,\mu\text{mol}\,\text{m}^{-2}$ (relative difference of $0.02\,\%$), which suggests that the signal is smeared, but the total amount of measured $NO_2$ is almost conserved. A small difference is to be expected because of statistical noise from the Monte Carlo method (about $1.3\,\mu\text{mol}\,\text{m}^{-2}$ for 50000 photons).

AMFs calculated with the 3D-box AMFs module including the urban canopy (Fig. 6c) also show lower values over roads compared to AMFs calculated with 1D-layer AMFs. In addition, they also show lower values over areas with buildings. As a consequence, the SCDs shown in Fig. 6f are lower over regions with many buildings. On average, $SCD_{3D-UC}$ are 12% lower than SCDs without buildings. The standard deviation of the difference between $SCD_{3D-UC}$ and $SCD_{3D}$ is $12.3\,\mu\text{mol}\,\text{m}^{-2}$ for the whole domain. Not including the urban canopy would therefore underestimate VCDs by 12% and would add a source of noise in the image of about 5% for this particular scenario with rather low buildings. In areas with background $NO_2$ and no buildings (e.g., lower left region) $SCD_{1D}$, $SCD_{3D}$ and $SCD_{3D-UC}$ closely agree (Fig. 7c).

The results presented above were obtained for the special situation where viewing and illumination directions were along the same east-west direction. In this case, the smearing effects are most prominent along this axis but relatively small in perpendicular direction. Here, we also analyze the situation where the sun is in the south and thus viewing and illumination directions are perpendicular to each other. As shown in Fig. 9, the results are generally similar but building shadows and correspondingly reduced SCDs are now found to the north instead of the east of the buildings. Furthermore, the SCDs tend to be lower because the main optical path and the corresponding smearing is both N-S (sun to ground pixel) and E-W-oriented (ground pixel to the instrument). The mean difference between $SCD_{3D-UC}$ and $SCD_{3D}$ is 12% and the standard deviation is 13%.

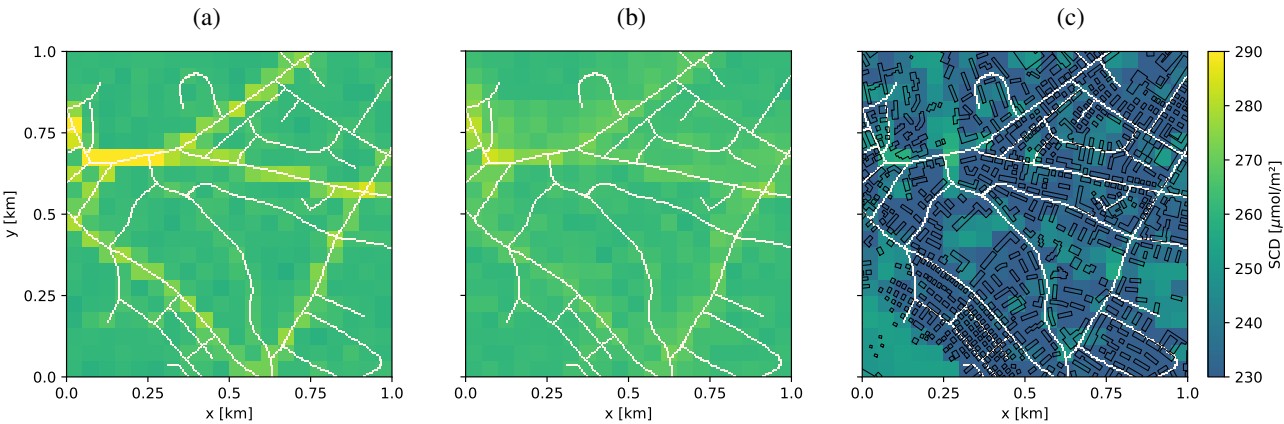

**Figure 9.** SCDs for simulations with a SAA of $0°$ with 1D-layer AMFs (a), 3D-box AMFs simulation without (b) and with (c) buildings. The roads are drawn in white and the building contours in black for the simulation with buildings.

### 3.3.2 3D effects at higher spatial resolution

To investigate the reduced SCDs over buildings observed in the former paragraphs, we analyse the spatial distribution of SCDs within the 50 m x 50 m pixel in more details. For this purpose, we ran additional simulations at higher spatial resolution (5 m x 5 m) for a 100 m x 100 m sub-domain located at x=600 m and y=0 m (lower-left corner of the sub-domain) of the original domain. SCDs were calculated from 3D-box AMFs with and without buildings (Fig. 10a and b).

At this resolution, we can better resolve the spatial distribution of $NO_2$ and SCDs remain high above roads. Since we can also resolve individual buildings and their shadows, the simulations makes it possible to investigate how buildings reduce SCDs.

At 5 m resolution, we distinguish four types of ground pixels. (1) The ground pixel is on the surface and the geometric path to the sun and the instrument is not obscured by buildings. In this case, the 3D-box AMFs are only affected by buildings through multiple scattering either increasing the AMF when buildings reflect photons towards the ground pixel or reducing the AMF when buildings block photons from reaching the ground pixel. In our example, the effect results in a small but hardly visible reduction of AMFs (see Fig. 10c).

(2) The reflecting point of the geometric path is located on top of a building. The case is similar to the first case, but 3D-box AMFs are smaller, because photons cannot reach the ground. Since we assume that a priori VCDs are fixed regardless of the presence of a building, SCDs are reduced above buildings. However, the effect is very small because only about 3 % of the total VCD is below a 10 m building.

(3) In the third case, the ground pixel is located on the sunlit side of a building, but the direct path to the instrument is blocked by the buildings. Since the VZAs of the simulated instrument are very small, we do not find these cases in our example with rather small buildings.

(4) In the final case, the ground pixel is located in the shadow of the buildings blocking the direct path towards the sun. In this case, photons can only reach the instrument after multiple scattering or simple atmospheric scattering directly into the direction of the instrument, which drastically reduces the 3D-box AMFs near the surface. As a result, SCDs are significantly lower in the shadows of buildings (Fig. 10b). For example, SCDs are about 35% lower when increased ground $NO_2$ concentrations (e.g., road) are located in the shadow of a building.

We can therefore conclude that the reduction of SCDs over buildings (Fig. 7c) is mainly caused by the building shielding effect (case 4), while building height has a minor effect on the SCDs (case 2) in our study case with rather small buildings.

### 3.3.3 The retrieval of vertical column densities

In imaging remote sensing, $NO_2$ VCDs are retrieved from SCDs using AMFs. Here $SCD_{3D-UC}$ are considered as the "true" SCDs measured by an airborne imaging spectrometer and VCDs were calculated using either 1D-layer AMFs or 3D-box AMFs without buildings (Fig. 11). The VCDs computed with 1D-layer AMFs fails to correct for the spatial smoothing induced by the complex 3D optical path of the photons (Fig. 11b). In additions, the effects of buildings are not corrected resulting in additional noise introduced by shielding effect of the buildings and significantly lower VCDs with a field average of 90.1 $\mu mol\,m^{-2}$

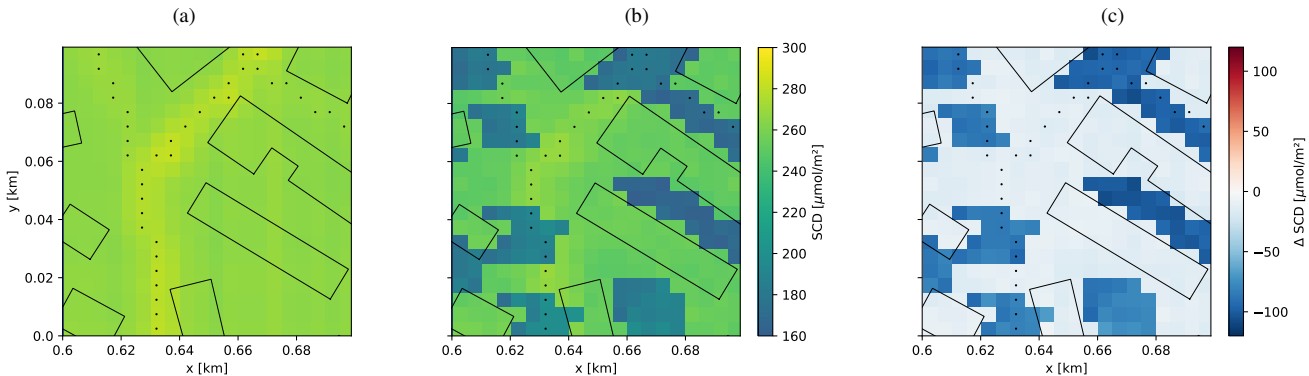

**Figure 10.** SCDs over a small sub-domain computed from high-resolution 3D-box AMFs simulated without (a) and with (b) buildings and the difference between both (c). In the simulation the sun was located in the west at a SZA of 60°. The albedo of roofs, walls and streets was set to 0.1. Roads are included as black dots and building contours as black lines.

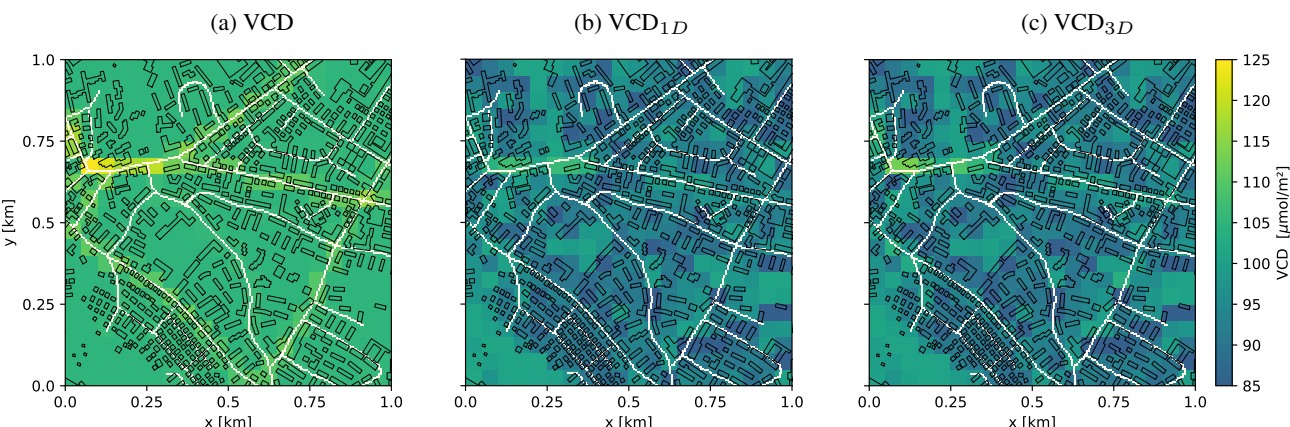

**Figure 11.** (a) True VCD field and VCDs computed from SCDs using 3D-box AMFs and buildings assuming (b) 1D-layer AMFs and (c) 3D-box AMFs without buildings. The scenario uses a SAA of 90°.

compared to the $108.4\,\mu\mathrm{mol\,m^{-2}}$ of the true VCDs. When VCDs are computed using 3D-box AMFs without buildings, the spatial smearing is corrected but the noise component and lower VCDs (mean: $91.4\,\mu\mathrm{mol\,m^{-2}}$) would remain in the retrieved VCDs.

## 3.4 Application to real observations

The codes developed for this study can also be applied to real observations, for example, to the campaigns conducted with APEX imaging spectrometer. A major challenge is to obtain the required input data. 3D building data are available for many cities, but albedos for ground, roof and walls are generally not available. In addition, to compute the total AMFs, realistic 3D $NO_2$ fields from a building-resolving dispersion model are required, which requires high-resolution emission inventories and additional model development, because most building-resolving models are not optimized for providing realistic vertical distributions of trace gases or cannot not be applied to a full city at high resolution (Berchet et al., 2017).

To minimize 3D effects when using 1D-layer AMFs, it would be recommendable to obtain the airborne spectrometer measurement around local noon when the SZA is lowest and avoid large viewing zenith angles. However, around noon turbulent atmospheric mixing will be strong and the $NO_2$ distributions would be smoothed as well.

The computation of 3D-box AMFs with buildings is computationally quite expensive, but still manageable for current airborne campaigns. For example, the computation of the 3D-box AMF field for a single APEX pixel (e.g. on Fig. 6f) takes about 280 s on a single core of our Linux machine (Intel(R) Xeon(R) W-2175 CPU @ 2.50GHz). Processing a full campaign consisting of about 100'000 pixels takes about 23 days using all 14 cores on the system. However, simulating AMFs for an APEX campaign would not require simulation for a 1 km x 1 km domain. Nonetheless, computing 3D-box AMFs is significantly more expensive than computing 1D-layer AMFs and reducing computation time, for example, by finding suitable parametrizations using machine learning, would make it possible to calculate the 3D-box AMFs on smaller hardware, for larger campaigns or to run simulations with more details and at higher spatial resolution.

## 4 Conclusions

Airborne imaging spectrometers are increasingly used for high-resolution mapping of $NO_2$ concentrations in cities. The $NO_2$ maps obtained in this way were usually found to be rather smooth and seemingly inconsistent with the much more rapidly varying near-surface $NO_2$ concentration fields seen, for example, in city-scale dispersion model simulations. The observed difference may partly be explained by atmospheric mixing more strongly affecting total columns than near-surface concentrations, but could also be caused by complex 3D radiative transfer effects in cities. To study the latter point, we implemented an urban canopy module into the 3D MYSTIC solver of the libRadtran radiative transfer model. We set up a case study for a 1 km x 1 km domain in Zurich, for which 3D-box AMFs and $NO_2$ slant column densities were computed for a realistic field of $NO_2$ concentrations.

Our case study shows that the footprint of a single observation is only partly located over the observed ground pixel and that there is a 'tail' in the direction of the main optical path. In the presented simulations with a SZA of 60° and a 50 m x 50 m resolution, about 50% of the sensitivity is located outside the ground pixel for a nearly-nadir viewing instrument. Only a small but not negligible amount of photons are from outside the main optical path, with 19% for a simulation without aerosols and without buildings and 24% for a simulation without aerosols and with buildings. The effect becomes more important when aerosols are included with 25% and 32% of the sensitivity located outside the main optical path for scenarios without and with

buildings, respectively. The footprint fine structure is further modified with the presence of buildings, but the general shape is conserved.

The 3D radiative transfer simulations show that 3D effects introduce significant spatial smearing of high $NO_2$ concentrations as for example over roads, which 1D-layer AMFs do not include. This results in increasing SCDs when roads are parallel to the main optical path and decreasing SCDs otherwise. When buildings are included, $NO_2$ SCDs are generally lower due to the shielding effect of buildings. The buildings also introduce a variability in the SCD field with a standard deviation of $12.9\,\mu mol\,m^{-2}$ (5.5%) that would show up as additional noise component of airborne imagers. The magnitude is however slightly smaller than the current $NO_2$ SCD uncertainty (about $20\,\mu mol\,m^{-2}$ for the APEX instrument), but could be noticeable for instruments dedicated to $NO_2$ mapping that have lower SCD uncertainties. We also applied 1D-layer and 3D-box AMFs without building to SCDs computed with 3D-box AMFs with buildings showing that 3D radiative transfer simulations are required to correct the smearing effect and that buildings are required to avoid an underestimation of the VCDs.

Generalizing our results to others cities is challenging, because many relevant parameters such as building shapes, surface reflectances and a priori $NO_2$ distribution vary strongly between different cities. In our case study, we used a surface reflectance of 0.1, which is a realistic value for Zurich but not necessarily for other cities. In general, a higher surface reflectance of the observed ground pixel implies less atmospheric scattering and a higher sensitivity of the instrument to the main optical path and higher albedo of neighbouring pixels increases the sensitivity to this neighbouring pixel. In this study, the simulations were conducted at $490\,nm$, which corresponds to the center of the fitting window used for $NO_2$ retrieval from the APEX airborne spectrometer. At shorter wavelength, used by other instruments, scattering increases, which decreases the instrument sensitivity to the main optical path. The footprint simulated with a wavelength of $420\,nm$ (without buildings) shows the increase in scattering and the sensitivity to neighbouring pixels, as 56% of the sensitivity is located outside of the ground pixel.

In conclusion, our case study demonstrates that 3D effects explain the smooth $NO_2$ field observed by airborne imaging spectrometers, at least partly. Atmospheric mixing can still result in additional smoothing that has not been studied here. Furthermore, buildings reduce SCDs due to light shielding effect of buildings and add an additional noise component that is difficult to generalize due to the complexity and the heterogeneity of the buildings. The smearing in sun direction can result in features in the maps that are difficult to interpret when the sun position is not known. 3D radiative transfer effects therefore need to be considered when studying $NO_2$ maps obtained from airborne imagers and might become relevant with future $NO_2$ satellite instruments that measure $NO_2$ at spatial resolutions down to $2\,km$.

*Code availability.* The libRadtran package including the 1D version of MYSTIC is freely available on www.libradtran.org, the 3D MYSTIC code is available upon request to Claudia Emde (claudia.emde@lmu.de). The Python script used to create a triangular mesh from ESRI shapefiles is available in the supplement and the other used codes are available upon request to the corresponding author.

*Data availability.* The data used for this study is available on https://doi.org/10.5281/zenodo.5519616.

*Author contributions.* MS designed and implemented the urban canopy module, designed and simulated the 3D scenarios and wrote the manuscript with input from all co-authors. FJ designed and implemented the urban canopy module together with MS and CE. DB, BB and AB provided critical feedback to the study and reviewed the manuscript, GK supervised the study, designed together with MS the case studies and reviewed the manuscript.

*Competing interests.* The authors declare that they have no conflict of interest.

*Acknowledgements.* This study was conducted as part of the HighNOCS project funded by the Swiss National Science Foundation (SNSF) under project number 172533. We obtained building data from the Swiss Federal Office of Topography (swisstopo) and the traffic emission inventories from the City of Zurich.

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
