# Peer review of "Impact of 3D radiative transfer on airborne NO2 imaging remote sensing over cities with buildings"

_Atmospheric Measurement Techniques, 2021_

## Referee Comment (RC1)

**Review of: Impact of 3D radiative transfer on airborne NO2 imaging remote sensing over cities with buildings (Schwaerzel et al., 2021)**

The manuscript discusses the study of 3D radiative transfer effects on $NO_2$ retrievals from airborne hyperspectral imaging systems. Also the impact of buildings on the light path is addressed. These effects are demonstrated in detail based on a 1 km x 1 km built-up region in Zurich and a realistic synthetic $NO_2$ field. The scientific content of the paper fits well within the scope of AMT and is valuable for (future) (NO2) retrievals from airborne and high resolution spaceborne observations. The manuscript is well-written and generally well-structured. Therefore, I highly recommend its publication in AMT. However, a number of revisions (detailed below) need to be conducted in the paper before publication.

**General comments**

-The studied 3D effects are indeed a very relevant problem in case of airborne/high resolution spaceborne trace gas retrievals over urban areas. A comment I had on the previous study of Schwaerzel et al., 2020 and also here is that I'm eager to see the impact on the VCD of using 3D BOX-AMFs (with/without building layer) instead of 1D layer-AMFs on a real-world airborne data set. Just as demonstrated with the synthetic data, a sample of an APEX data set acquired over Zurich could be used for this.

-Sect 3.3: I suggest to add a figure and description where you compute back the VCDs, e.g. based on the SCD-3D (considered that this is the typical smoothed NO2 field observed with an airborne imager) and the AMF-1D. This could demonstrate the impact of smoothing on the retrieved VCD (when compared to the 'true' simulated NO2 field of Fig. 2c) when not considering 3D effects. Same could be done by computing the VCD based on SCD-3D-UC and AMF-3D to demonstrate the impact of adding/neglecting the urban canopy. This would help the interpretation and avoid that a non-careful reader could have the impression that the 3D case leads to a more smoothed NO2 field when looking at the SCDs in Figure 6.

-p.2, l.51: This work is building further on an earlier work, i.e. Schwaerzel et al. (2020), and there is clearly an overlap. Please add here explicitly in which way this new study differentiates from the previous study. The full study is not only focusing on the addition of the urban canopy.

-p.5, l.125: Although each building surface can have its own albedo, for reasons of simplicity everything is given the same albedo of 0.1 in the study discussed in 3.3. It is somewhat confusing as sometimes "building shadows" are mentioned in the discussion (pointing to a different albedo). However, I think the authors refer with 'building shadows" rather to areas of the surface that cannot be 'seen' or are 'shielded' due to building obstruction of the lightpath. If this is the case, I would suggest to be more careful with the use of "shadows" or "shadowing effects" in the further discussion and maybe mention it as a 'shielding' effect. I also would explicitly repeat the made assumptions on the albedo in Sect. 3.3.

**Minor comments**

-Title: "Cities with buildings" sounds a bit weird. Maybe replace by "built-up areas"?

-p.2, l.28: You might consider adding the following reference : Vlemmix, T., Ge, X. (., de Goeij, B. T. G., van der Wal, L. F., Otter, G. C. J., Stammes, P., Wang, P., Merlaud, A., Schüttemeyer, D., Meier, A. C., Veefkind, J. P., and Levelt, P. F.: Retrieval of tropospheric $NO_2$ columns over Berlin from high-resolution airborne observations with the spectrolite breadboard instrument, Atmos. Meas. Tech. Discuss. [preprint], https://doi.org/10.5194/amt-2017-257, in review, 2017.

-p.4, l.105: Not clear what the difference is between 'material' and 'material type'. Please specify.

-p.5, l.125: Albedo typically has a strong impact on the AMF. Since the focus of this study is on adding the urban canopy (and its impact), it would be useful for further studies to do a sensitivity test to show the impact on the AMF of different typical roof types (even if the focus of the study here is not specifically on the building albedo).

-p.7, l.155: I assume this is done for each along-track pixel?

-p.9, l.205: "The slight increase in the column AMFs in the right is induced by scattering events in the right side of the domain appearing on the left of the domain due to circular boundaries" → This effect is not well understood.

-p.11, l.251: AMFs calculated with 3D-box AMFs but without buildings (Figure 6b) are lower over the roads and slightly larger just aside the roads." → it would help the reader to repeat why this is the case.

-p.12, l. 268: "Not including the urban canopy would therefore underestimate VCDs by 12%..." → I first thought to elaborate a bit on this in the conclusion. However, I think it is difficult to generalize, depending on the complexity of the urban canopy and also due to the fact that the bulk of the NO2 can be in elevated layers in real-world conditions which would reduce the impact of adding the UC.

-p.15, 322: "When buildings are included, NO2 SCDs are generally lower due to the shadows of building" → As you assume a same surface reflectance in the study I assume this is rather due to the blocking of the lighpath?

-Conclusions: Some suggestions would be helpful for future airborne/spaceborne missions and/or instrument design to reduce 3D effects over urban areas, e.g. to operate close to local noon and maybe operate with small VZA, or to put some thresholds on SZA and VZA? The latter is of course a trade-off with the amount of data than can be acquired during an overpass.

**Technical corrections**

-p.2, l.31: to → towards the instrument

-p.2,l.40: is NO2 maps → is that NO2 maps

-p.7,l.180: Fig.2c → Fig. 2c (Note as well that references to figures are not always consistent throughout the manuscript. Sometimes written as Figure 1, sometimes as Fig. 1)

-p.15, 314: in the direction **of**

-p.16, 326: Two times "have"

-Fig. 8: This caption is not self-explanatory. Also in the text the figure is not clearly explained.

-Please have a close look at the supplement to correct for typos, e.g. p.2, l. 55: should be 'emission'; add space between 15 and [; 'this' parameters → 'these' parameters, etc.

---

## Author Comment (AC1)

**Impact of 3D radiative transfer on airborne NO2 imaging remote sensing over cities with buildings**

Marc Schwaerzel et al. 2021

**Response to the Reviewer's Comments**

We thank Frederik Tack for his positive comments, critical assessment and useful points to improve the quality of our paper. In the following we address his concerns point by point. Changes in the paper are shown in blue. We hope we clarified all concerns and that the revised manuscript has improved.

**5 Reviewer 3**

10

**Reviewer Point P 3.1** — The studied 3D effects are indeed a very relevant problem in case of airborne/high resolution spaceborne trace gas retrievals over urban areas. A comment I had on the previous study of Schwaerzel et al., 2020 and also here is that I'm eager to see the impact on the VCD of using 3D BOX-AMFs (with/without building layer) instead of 1D layer-AMFs on a real-world airborne data set. Just as demonstrated with the synthetic data, a sample of an APEX data set acquired over Zurich could be used for this.

**Reply**: We agree that the application to real observations is important and we are currently in the process to apply 3D-box AMFs to real data from an APEX campaign over Zurich and eventually Munich. However, this step also requires additional work, for example, setting up city-scale dispersion models to have suitable 3D NO2 fields. Since this additional work would make the manuscript substantially larger, we decided against adding it here.

- 15 Reviewer Point P 3.2 I suggest to add a figure and description where you compute back the VCDs, e.g. based on the SCD-3D (considered that this is the typical smoothed NO2 field observed with an airborne imager) and the AMF-1D. This could demonstrate the impact of smoothing on the retrieved VCD (when compared to the 'true' simulated NO2 field of Fig. 2c) when not considering 3D effects. Same could be done by computing the VCD based on SCD-3D-UC and AMF-3D to demonstrate the impact of adding/neglecting the urban canopy. This would
- 20 help the interpretation and avoid that a non-careful reader could have the impression that the 3D case leads to a more smoothed NO2 field when looking at the SCDs in Figure 6.

**Reply**: This is a good suggestion, we added a new section addressing the computation/retrieving of VCDs from (simulated) SCDs that include 3D effects and buildings. We added the following three figures showing VCD fields calculated from the

SCDs calculated with 3D-box AMFs with buildings. Additionally we added a figure showing the VCDs retrived from SCDs

25 calculated with 3D-box AMFs without buildings to the supplement. Figure 1 compares VCDs calculated from the new assumed "true" SCD3D-UC combined with AMFs from (left) 3D-box AMFs accounting for buildings (middle) 1D-layer AMFs and (right) 3D-box AMFs ignoring buildings.

Figure 1. VCDs for a simulation with SAA of 90° with 3D-box AMFs simulation (left) with and (right) without buildings, and with (middle) 1D-layer AMFs.

In the main text we added a subsection containing the following text:

In imaging remote sensing, NO2 VCDs are retrieved from SCDs using AMFs. Here  $SCD_{3D-UC}$  are considered as the "true" SCDs measured by an airborne imaging spectrometer and VCDs were calculated using either 1D-layer AMFs or 3D-box AMFs without buildings (Fig. 11). The VCDs computed with 1D-layer AMFs fails to correct for the spatial smoothing induced by the complex 3D optical path of the photons (Fig. 11b). In additions, the effects of buildings are not corrected resulting in additional noise introduced by shielding effect of the buildings and significantly lower VCDs with a field average of 90.1 µmol m-2 compared to the 108.4 µmol m-2 of the true VCDs. When VCDs are computed using 3D-box AMFs without buildings, the spatial smearing is corrected but the noise component and lower VCDs (mean: 91.4 µmol m-2) would remain in the retrieved VCDs.

**Reviewer Point P 3.3** — This work is building further on an earlier work, i.e. Schwaerzel et al.(2020), and there is clearly an overlap. Please add here explicitly in which way this new study differentiates from the previous study. The full study is not only focusing on the addition of the urban canopy.

Reply: We clarified this point adding the following sentence:

The study builds on the work by Schwaerzel et al. (2020), where we highlighted the importance of 3D RT effects on trace gas remote sensing for ground-based and airborne instruments.

35

40

30

**Reviewer Point P 3.4** — Although each building surface can have its own albedo, for reasons of simplicity everything is given the same albedo of 0.1 in the study discussed in 3.3. It is somewhat confusing as sometimes 45 "building shadows" are mentioned in the discussion (pointing to a different albedo). However, I think the authors refer with 'building shadows' rather to areas of the surface that cannot be 'seen'or are 'shielded' due to building obstruction of the lightpath. If this is the case, I would suggest to be more careful with the use of "shadows" or "shadowing effects" in the further discussion and maybe mention it as a 'shielding' effect. I also would explicitly repeat the made assumptions on the albedo in Sect. 3.3. 50

**Reply**: The presence of building shadows does not change the albedo of a surface, because the albedo is a property independent of illumination conditions. We have used the term "building shadowing" where the ground pixel is located in the building shadow and, therefore, the direct optical path towards the sun is shielded by the building. However, we noticed that the term "shadowing" is used in BRDF products to describe ground pixels where objects cast shadows. Since this can lead to confusion, we have replaced the term "building shadowing" with "building shielding" in the manuscript.

55

**Reviewer Point P 3.5** — Title: "Cities with buildings" sounds a bit weird. Maybe replace by "built-up areas"? **Reply**: We understand your point, as we had the same discussions on the title internally but decided on using "cities with buildings" because it emphasises that buildings have not yet been taken into account in airborne  $NO_2$  remote sensing.

- Reviewer Point P 3.6 p.2, l.28: You might consider adding the following reference : Vlemmix, T., Ge, X. 60 (., de Goeij, B. T. G., van der Wal, L. F., Otter, G. C. J., Stammes, P., Wang, P., Merlaud, A., Schüttemeyer, D., Meier, A. C., Veefkind, J. P., and Levelt, P. F.: Retrieval of tropospheric NO2 columns over Berlin from highresolution airborne observations with the spectrolite breadboard instrument, Atmos. Meas. Tech. Discuss. [preprint], https://doi.org/10.5194/amt-2017-257, in review, 2017.
- **Reply**: We added the citation. 65

**Reviewer Point P 3.7** — -p.4, 1.105: Not clear what the difference is between 'material' and 'material type'. Please specify

**Reply**: We have rewritten the description of the NetCDF file as follows:

The triangular mesh is stored in a NetCDF file readable by MYSTIC. An example file layout is shown 70 in the supplement. The file contains the variable vertices (shape:  $N_v \times 3$ , type: double), which is a list of x, y and z coordinates. A mesh of  $N_t$  triangles is build from these vertices using the triangles variable (shape:  $N_t \times 3$ , type: int) by storing the indices of the 3 vertices that create the triangles. The variable materials\_of\_triangles (shape:  $N_t$ , type: int) is used to assign each triangle the index of a material type. The material types are defined using the variables material\_type (shape:  $N_m$ , type: string), material\_albedo (shape:  $N_m$ , type: double) and temperature\_of\_triangle (shape:  $N_m$ , type: double) to assign a name, albedo and temperature to each material.

75

80

85

**Reviewer Point P 3.8** — p.5, l.125: Albedo typically has a strong impact on the AMF. Since the focus of this study is on adding the urban canopy (and its impact), it would be useful for further studies to do a sensitivity test to show the impact on the AMF of different typical roof types (even if the focus of the study here is not specifically on the building albedo).

**Reply**: We agree that conducting sensitivity studies for varying roof albedos (and other parameters) would be interesting further studies. We partially showed the effect magnitude of a different type of albedo in the supplement for a roof albedo of 0.2. However, we would like to keep the examples simple here to introduce the general concept and leave detailed sensitivity studies for later work.

**Reviewer Point P 3.9** — p.7, l.155: I assume this is done for each along-track pixel?

Reply: Yes, each along-track pixel is an average of 10 discrete steps of 5 m (i.e. 10 instrument positions).

90 Reviewer Point P 3.10 — p.9, l.205: "The slight increase in the column AMFs in the right is induced by scattering events in the right side of the domain appearing on the left of the domain due to circular boundaries" This effect is not well understood.

**Reply**: Since MYSTIC uses circular boundary conditions, photons leaving the model domain on the left will enter the domain on the right. As a result, column AMFs on the right side of the ground pixel can be increased when the main optical passes overhead in the upper atmosphere for small domains and large SZAs. This model artefact was visible in an early version of the figure. In the current version the artefact cannot be seen anymore, because we only integrate up to 0.4 km. We removed the sentence in the revised version to avoid confusion.

Reviewer Point P 3.11 — p.11, l.251: AMFs calculated with 3D-box AMFs but without buildings (Figure 6b)
are lower over the roads and slightly larger just aside the roads." -> it would help the reader to repeat why this is the case.

Reply: We extended the sentence with the following:

, because the 3D-optical path crosses neighbouring columns with decreased or increased concentrations, respectively.

105 Reviewer Point P 3.12 — p.12, l. 268: "Not including the urban canopy would therefore underestimate VCDs by 12%..."I first thought to elaborate a bit on this in the conclusion. However, I think it is difficult to generalize, depending on the complexity of the urban canopy and also due to the fact that the bulk of the NO2 can be in elevated layers in real-world conditions which would reduce the impact of adding the UC.

**Reply**: We agree that a generalisation is difficult because it depends on various factors (urban canopy structure, re-110 flectances,  $3D NO_2$  distribution). We briefly discuss the limitation of our scenario in the conclusions.

Generalizing our results to others cities is challenging, because many relevant parameters such as building shapes, surface reflectances and a priori  $NO_2$  distribution vary strongly between different cities. In our case study, we used a surface reflectance of 0.1, which is a realistic value for Zurich but not necessarily for other cities. In general, a higher surface reflectance of the observed ground pixel implies less atmospheric scattering and a higher sensitivity of the instrument to the main optical path and higher albedo of neighbouring pixels increases the sensitivity to this neighbouring pixel.

115

125

**Reviewer Point P 3.13** — p.15, 322: "When buildings are included, NO2 SCDs are generally lower due to the shadows of building" As you assume a same surface reflectance in the study I assume this is rather due to the blocking of the lighpath?

120 **Reply**: Yes (see our reply to point 3.4).

**Reviewer Point P 3.14** — Conclusions: Some suggestions would be helpful for future airborne/spaceborne missions and/or instrument design to reduce 3D effects over urban areas, e.g. to operate close to local noon and maybe operate with small VZA, or to put some thresholds on SZA and VZA? The latter is of course a trade-off with the amount of data than can be acquired during an overpass.

**Reply**: Yes, this is a relevant point. We added a new section 3.4 where we discuss application to real observations and give some recommendations for future campaigns.

The codes developed for this study can also be applied to real observations, for example, to the campaigns conducted with APEX imaging spectrometer. A major challenge is to obtain the required input data. 3D building data are available for many cities, but albedos for ground, roof and walls are generally not available. In addition, realistic 3D NO2 fields from a building-resolving dispersion model are required to compute the total AMFs, which requires high-resolution emission inventories and additional model development, because most building-resolving models are not optimized for providing realistic vertical distributions of trace gases or cannot not be applied to a full city at high resolution (Berchet et al., 2017).

135 To minimize 3D effects when using 1D-layer AMFs, it would be recommendable to obtain the airborne spectrometer measurement around local noon when the SZA is lowest and avoid large viewing zenith angles. However, around noon turbulent atmospheric mixing will be strong and the NO2 distributions would be smoothed as well.

**Reviewer Point P 3.15** — p.2, l.31: to -> towards the instrument-

140 **Reply**: We corrected this point.

130

Reviewer Point P 3.16 — p.2,1.40: is NO2 maps -> is that NO2 maps-

Reply: We corrected this point.

**Reviewer Point P 3.17** — p.7,l.180: Fig.2c -> Fig. 2c

Reply: We corrected this point.

145 Reviewer Point P 3.18 — p.7,l.180: Fig.2c -> Fig. 2c (Note as well that references to figures are not always consistent throughout the manuscript. Sometimes written as Figure 1, sometimes as Fig. 1)

**Reply**: Thank you, we replaced all "Figure" located in the middle of a sentence by "Fig." as required by the AMT style guideline.

Reviewer Point P 3.19 — -p.15, 314: in the direction of -p.16, 326: Two times "have"

150 **Reply**: We corrected these mistake.

**Reviewer Point P 3.20** — -Fig. 8: This caption is not self-explanatory. Also in the text the figure is not clearly explained.

Reply: We clarified the reference to the figure in the text and modified the caption as following:

155

**Reviewer Point P 3.21** — -Please have a close look at the supplement to correct for typos, e.g. p.2, l. 55: should be 'emission'; add space between 15 and [; 'this' parameters 'these' parameters, etc

**Reply**: We have revised the language and grammar in the supplement and changed "immission" to "concentration" as we refer to a measurable concentration and emissions here.

**160 References**

- Berchet, A., Zink, K., Muller, C., Oettl, D., Brunner, J., Emmenegger, L., and Brunner, D.: A cost-effective method for simulating city-wide air flow and pollutant dispersion at building resolving scale, Atmospheric Environment, 158, 181–196, https://doi.org/https://doi.org/10.1016/j.atmosenv.2017.03.030, 2017.
- Schwaerzel, M., Emde, C., Brunner, D., Morales, R., Wagner, T., Berne, A., Buchmann, B., and Kuhlmann, G.: Threedimensional radiative transfer effects on airborne and ground-based trace gas remote sensing, Atmospheric Measurement Techniques, 13, 4277–4293, https://doi.org/10.5194/amt-13-4277-2020, 2020.

---

## Author Comment (AC2)

**Impact of 3D radiative transfer on airborne NO2 imaging remote sensing over cities with buildings**

Marc Schwaerzel et al. 2021

**Response to the Reviewer's Comments**

We thank Reviewer 2 for his/her positive comments, critical assessment and useful points to improve the quality of our paper. In the following we address his/her concerns point by point. Changes in the paper are shown in blue. We hope we clarified all concerns and that the revised manuscript has improved.

5 ## Reviewer 2

**Reviewer Point P 2.1** — This paper looks to address the 3-d radiative transfer effects of urban landscapes on NO2 retrievals. The authors use monte carlo simulations from MYSTIC for a simplified urban landscape in Zurich to examine the impacts of buildings AMF calculations (so called 3D-box AMF). This is a very interesting study with significant implications to airborne retrievals. The only part the authors don't seem to address is how they would

10 account for such 3-d effects in actual airborne retrievals. Many of the assumptions they make in this study would likely not be applicable to real world retrievals. Maybe this will come in a later paper, but it would make the paper stronger to explain how this could be translated into actual retrievals.

**Reply**: We understand the reviewer's point on the lack of information concerning a real application. We have added a subsection discussing the application to real data. This matter, will be the core of a further study, where we will apply

15 3D-box AMFs to real APEX data.

> The codes developed for this study can also be applied to real observations, for example, to the campaigns conducted with APEX imaging spectrometer. A major challenge is to obtain the required input data. 3D building data are available for many cities, but albedos for ground, roof and walls are generally not available. In addition, realistic 3D $NO_2$ fields from a building-resolving dispersion model are required
>
> 20 to compute the total AMFs, which requires high-resolution emission inventories and additional model development, because most building-resolving models are not optimized for providing realistic vertical distributions of trace gases or cannot not be applied to a full city at high resolution (Berchet et al., 2017).

To minimize 3D effects when using 1D-layer AMFs, it would be recommendable to obtain the airborne spectrometer measurement around local noon when the SZA is lowest and avoid large viewing zenith angles. However, around noon turbulent atmospheric mixing will be strong and the $NO_2$ distributions would be smoothed as well.

The computation of 3D-box AMFs with buildings is computationally quite expensive, but still manageable for current airborne campaigns. For example, the computation of the 3D-box AMF field for a single APEX pixel (e.g. on Fig. 6f) takes about $280\,s$ on a single core of our Linux machine (Intel(R) Xeon(R) W-2175 CPU @ 2.50GHz). Processing a full campaign consisting of about 100'000 pixels takes about 23 days using all 14 cores on the system. However, simulating AMFs for an APEX campaign would not require simulation for a $1\,km$ x $1\,km$ domain. Nonetheless, computing 3D-box AMFs is significantly more expensive than computing 1D-layer AMFs and reducing computation time, for example, by finding suitable parametrizations using machine learning, would make it possible to calculate the 3D-box AMFs on smaller hardware, to larger campaigns or to run simulations with more details and at higher spatial resolution.

**Reviewer Point P 2.2** — The authors note that 50% of the NO2 sensitivity is from outside the ground pixel for a nadir viewing geometry. They also note that the urban canopy module in MYSTIC currently only supports Lambertian reflections. Could the authors elaborate on how the Lambertian assumption would effect their results? Would it be a safe assumption that accounting for specular reflection you would have less light scattering in from outside the ground pixel?

**Reply**: The sensitivity to neighboring pixels depends strongly on the reflectance properties of the neighboring pixels. Considering non-Lambertian reflection would increase the complexity of the analysis. If all neighboring pixels are specular reflecting, multiple scattering is required for photons to reach the instrument (except if the direct light path crosses the light-of-sight of the instrument), while only single scattering is required for Lambertian surfaces. As a result, sensitivity would decrease for specular reflection. For real application, surface reflectance would be best described by BRDF functions. We added the following line in the manuscript:

For an even more realistic description of the surface reflectance for real applications, the BRDF reflection function would be a well suited method.

**Reviewer Point P 2.3** — The simple assumptions about albedo seem to not be very realistic. It would be nice to do simulations with more reasonable albedos or at least provide some discussion on the possible errors from making these assumptions about the albedo.

**Reply**: We agree that albedo is important and has a big impact on AMFs. The albedos used in this study were taken from APEX measurements over our study area. While real albedos have some variations, the used values (0.1 and 0.2 for

roofs/walls and 0.1 for ground) are typical for this area of the city. However, it is true that albedos can vary for different areas and cities. We added the following paragraph in the conclusions to discuss this in more details:

> Generalizing our results to others cities is challenging, because many relevant parameters such as building shapes, surface reflectances and a priori $NO_2$ distribution vary strongly between different cities. In our case study, we used a surface reflectance of 0.1, which is a realistic value for Zurich but not necessarily for other cities. In general, a higher surface reflectance of the observed ground pixel implies less atmospheric scattering and a higher sensitivity of the instrument to the main optical path and higher albedo of neighbouring pixels increases the sensitivity to this neighbouring pixel.

**Reviewer Point P 2.4** — Pg 1 Ln 18. "by fuel combustion by traffic, heating systems...", suggest changing to "by fuel combustion, traffic, heating systems..."

**Reply**: We implemented the suggested change.

**Reviewer Point P 2.5** — Pg 2 Ln 53. Please spell out MYSTIC completely, the Monte carlo code for the phYSically correct Tracing of photons In Cloud atmospheres

**Reply**: We spell out MYSTIC completely now.

**Reviewer Point P 2.6** — Pg 8 Ln 186 "The reflectance was set to 0.1", please clarify that you are referring to surface reflectance, not top of atmosphere reflectance

**Reply**: We added "surface" before reflectance.

**Reviewer Point P 2.7** — Pg 15, Fig 10 It is curious that the building in the bottom right of the figure shows now shadowing effects despite the sun being to the west. Can you explain why this building is not effect by shadows effects. Even if it was in the shadow of the building to the west, one would still expect and impact on SCDs.

**Reply**: Excellent observation. We verified the simulation inputs and it appears that the building was missing, because we selected only buildings whose centroids are within the domain, whereas all buildings were included in the drawing of building contours. We run the simulation again including the building. Figure 10 was updated.

**Reviewer Point P 2.8** — Pg 15 Line 316 "with 19% and 24% for simulations without aerosols and without and with buildings, respectively". This statement is a bit confusing. I suspect you mean 19% from outside the optical path with no aerosol/no building and 24% from outside the optical path with no aerosol/with building, but it is not clear

**Reply**: Yes you are right. We clarified the sentence as following:

> Only a small but not negligible amount of photons are from outside the main optical path, with 19% for a simulation without aerosols and without buildings and 24% for a simulation without aerosols and with buildings.

**Reviewer Point P 2.9** — Pg 15 Line 317 should be "The effect becomes more.."

**Reply**: We corrected the typo.

**References**

Berchet, A., Zink, K., Muller, C., Oettl, D., Brunner, J., Emmenegger, L., and Brunner, D.: A cost-effective method for
95    simulating city-wide air flow and pollutant dispersion at building resolving scale, Atmospheric Environment, 158, 181–196,
https://doi.org/https://doi.org/10.1016/j.atmosenv.2017.03.030, 2017.

---

## Author Comment (AC3)

**Impact of 3D radiative transfer on airborne NO2 imaging remote sensing over cities with buildings**

Marc Schwaerzel et al. 2021

**Response to the Reviewer's Comments**

We thank Reviewer 3 for his/her positive comments, critical assessment and useful points to improve the quality of our paper. In the following we address his/her concerns point by point. Changes in the paper are shown in blue. We hope we clarified all concerns and that the revised manuscript has improved.
* * *
**5   Reviewer 1**

**Reviewer Point P 1.1** — This paper describes the influence of 3D radiative transfer and shadowing on airborne NO2 retrievals, using a case study over Zurich. The paper is well-organized, concise and easy to follow. I have a couple of general and specific comments. After those are addressed I would recommend it be published in AMT. The study results are interesting but I am not sure how the results can be transferred over to a practical application in retrievals;
10  however, it will be a good reference for the impacts of 3D radiative transfer on these kinds of measurements.

I find the description of the motivation to be unconvincing. The authors point to possible 3D effects as being responsible for the discrepancy between high resolution airborne NO2 maps over urban areas and city-scale urban models, and point to Figure S1 for an example. First, to drive home the need for this study, it would be good to see the motivating figures in the main paper instead of the supplement. To me, the observed and modeled NO2 maps
15  look so very different that I doubt the source of the differences is entirely or even primarily 3D radiative effects. I would suspect issues with the model like inaccuracies in mixed layer height, transport, emissions, and chemistry, or issues with surface reflectance and profile shapes in the retrievals. For instance, there are what look like three plumes in the southwest corner of the map which actually look quite well-represented. Why are these represented fairly well but other plumes are not? It may be that 3D effects are the main reason for observed/model discrepancies, but the
20  current example is unconvincing.

The study's motivation would be more convincing if: 1) The new calculations were actually applied to APEX data to calculate new AMFs, perhaps at the end of the paper, and the new maps showed better agreement with the model or improvements in resolution, or 2) a simulation were done using the GRAL NO2 columns to simulate airborne measurements that use 1D radiative transfer, and these simulations were found to show significant smearing. Even

25 if this is not possible, I would suggest leaving out the example figure, and be more nuanced in the motivation, i.e., along the lines of "we explore 3D effects as a possible contributor to smearing...". The later results will show whether or not they are significant.

**Reply**: We agree that difference between the APEX and GRAL $NO_2$ fields are not only caused 3D radiative transfer effects but also by limitations of the model, which is why it was not added to the main text. The cited conference presentation

30 actually lists some likely necessary model improvements. We have removed the figure from the supplement and instead published the presentation containing the figure on Zenodo (https://doi.org/10.5281/zenodo.5220909). The suggestion to apply the method to APEX data and to compare the map to a GRAMM-GRAL simulation is currently work in progress and will be the topic of future publications. We have revised the introduction keeping a stronger focus on how 3D radiative transfer effects might contribute to spatial smearing and also added a subsection on real application implications:

35 A strong indication for the importance of 3D radiative transfer effects is that $NO_2$ maps obtained from airborne imaging spectrometers over cities are spatially much smoother than one would expect from the instrument resolution and compared to maps obtained from high-resolution city-scale dispersion models, which show, for example, strong gradients in the $NO_2$ field along major roads (Kuhlmann et al., 2017).

**Reviewer Point P 1.2** — My second general comment is that it would be nice to see some discussion of the
40 practical implementation of these calculations. Is it too computationally intensive to use for actual campaign AMF calculations? How would a more realistic albedo field change the results?

**Reply**: We have added a new section in the paper showing the computation of VCDs from simulated SCDs and discussing the application to real observations as well as challenges (computational costs, realistic albedos and $NO_2$ fields, etc.):

The codes developed for this study can also be applied to real observations, for example, to the campaigns
45 conducted with APEX imaging spectrometer. A major challenge is to obtain the required input data. 3D building data are available for many cities, but albedos for ground, roof and walls are generally not available. In addition, realistic 3D $NO_2$ fields from a building-resolving dispersion model are required to compute the total AMFs, which requires high-resolution emission inventories and additional model development, because most building-resolving models are not optimized for providing realistic vertical
50 distributions of trace gases or cannot not be applied to a full city at high resolution (Berchet et al., 2017).

To minimize 3D effects when using 1D-layer AMFs, it would be recommendable to obtain the airborne spectrometer measurement around local noon when the SZA is lowest and avoid large viewing zenith angles. However, around noon turbulent atmospheric mixing will be strong and the $NO_2$ distributions would be smoothed as well.

55 The computation of 3D-box AMFs with buildings is computationally quite expensive, but still manageable for current airborne campaigns. For example, the computation of the 3D-box AMF field for a single

APEX pixel (e.g. on Fig. 6f) takes about 280 s on a single core of our Linux machine (Intel(R) Xeon(R) W-2175 CPU @ 2.50GHz). Processing a full campaign consisting of about 100'000 pixels takes about 23 days using all 14 cores on the system. However, simulating AMFs for an APEX campaign would not require simulation for a 1 km x 1 km domain. Nonetheless, computing 3D-box AMFs is significantly more expensive than computing 1D-layer AMFs and reducing computation time, for example, by finding suitable parametrizations using machine learning, would make it possible to calculate the 3D-box AMFs on smaller hardware, to larger campaigns or to run simulations with more details and at higher spatial resolution.

**Reviewer Point P 1.3** — Line 3: It's unlikely that the entire cause would be 3D radiative transfer effects. Should qualify with wording like "3D radiative transfer effects may contribute to this discrepancy due to...."

**Reply**: Yes you are right that the wording could be misleading. We modified the sentence as following:

This could partly be caused by 3D radiative transfer effects due to observation geometry, adjacency effects and effects of buildings.

**Reviewer Point P 1.4** — Line 152: Can you give an estimate (maybe in results section) about how much the results would change for shorter wavelength fitting windows? Shorter wavelengths are used in almost all other airborne and satellite instruments that measure NO2, so this would be most interesting for the majority of readers who do not use APEX data.

**Reply**: In general shorter wavelength increases scattering and decreases the instrument sensitivity. We simulated the footprint with a wavelength of 420 nm and without buildings to look at sensitivity distribution. It appears that 56% of the sensitivity is located outside the ground pixel. For simulations with buildings this number strongly depends on the buildings locations and the induced shadows and is therefore difficult to generalize. We added the following sentence in the conclusions:

In this study, the simulations were conducted at 490 nm, which corresponds to the center of the fitting window used for $NO_2$ retrieval from the APEX airborne spectrometer. At shorter wavelength, used by other instruments, scattering increases, which decreases the instrument sensitivity to the main optical path. The footprint simulated with a wavelength of 420 nm (without buildings) shows the increase in scattering and the sensitivity to neighbouring pixels, as 56% of the sensitivity is located outside of the ground pixel.

**Reviewer Point P 1.5** — Some mixing of verb tenses in paper. For example, in abstract say "We compute.." then next sentence says "We found..."

**Reply**: We checked verb tenses in the manuscript and fixed the mistakes.

**Reviewer Point P 1.6** — Line 63: I'm a bit confused at the wording of this sentence. Should it be "The model,
90   however, is able to . . . "

**Reply**: We modified the sentence as you suggested.

**References**

Berchet, A., Zink, K., Muller, C., Oettl, D., Brunner, J., Emmenegger, L., and Brunner, D.: A cost-effective method for simulating city-wide air flow and pollutant dispersion at building resolving scale, Atmospheric Environment, 158, 181–196, https://doi.org/https://doi.org/10.1016/j.atmosenv.2017.03.030, 2017.

Kuhlmann, G., Berchet, A., and Brunner, D.: High-resolution remote sensing and modelling of NO2 air pollution over the city of Zurich, in: 10th EARSeL SIG Imaging Spectroscopy Workshop 2017, Zurich, Switzerland, https://doi.org/10.5281/zenodo.5220909, 2017.

95

---

## Author Response (AR2)

Dear Lok Lamsal,

We want to thank you for your work and the nice final comment. We corrected the mistakes you pointed out in the manuscript.

Yours sincerely,
Marc Schwaerzel